# Omni×R: Evaluating Omni-modality Language Models on Reasoning across Modalities

**Lichang Chen**[12] *[,] **Hexiang Hu**[1], **Mingda Zhang**[1], **Yiwen Chen**[1], **Zifeng Wang**[1], **Yandong Li**[1],
**Pranav Shyam**[1], **Tianyi Zhou**[2], **Heng Huang**[2], **Ming-Hsuan Yang**[1], **Boqing Gong**[1]
Google DeepMind[1]; University of Maryland, College Park[2]

## Abstract

We introduce **Omni×R**, an evaluation suite designed to benchmark state-of-the-art Omni-modality Language Models (OLMs), such as GPT-4o and Gemini. Evaluating OLMs, which integrate multiple modalities such as text, vision, and audio, presents unique challenges. Particularly, the user message might often consist of multiple modalities, such that OLMs have to establish holistic understanding and reasoning across modalities to accomplish the task. Existing benchmarks are limited to single-modality or dual-modality tasks (e.g., image+text or video+text), overlooking comprehensive multi-modal assessments of model reasoning. To address this, Omni×R offers two evaluation variants: (1) Omni×R$_{\text{SYNTH}}$: a synthetic dataset generated automatically by translating text into multiple modalities—audio, images, video, and hybrids (`Omnify!`). (2) Omni×R$_{\text{REAL}}$: a real-world dataset, manually curated and annotated by experts, for evaluating cross-modal reasoning in natural settings. Omni×R presents a unique evaluation towards assessing OLMs over a diverse mix of modalities, such as a question that involves video, audio, and text, providing a rigorous cross-modal reasoning testbed than any existing benchmarks. Our experiments find that all state-of-the-art OLMs struggles with Omni×R questions that require integrating information from multiple modalities to answer. Further analysis highlight differences in reasoning behavior and underscoring the challenges of omni-modal AI alignment.

## 1 Introduction

Recent advances in Omni-modality Language Models (OLMs) (OpenAI, 2024b; Gemini-Team, 2024b) has pushed the boundaries of AI by enabling a more comprehensive understanding of real-world inputs across diverse modalities, e.g., text, vision, audio, (Lu et al., 2019; Gan et al., 2020; Akbari et al., 2021; Zellers et al., 2021) and generating outputs that are more aligned with human communications (Lu et al., 2024; Zhang et al., 2024; Gao et al., 2024).

However, the evaluation of these sophisticated OLMs presents unique challenges. While traditional benchmarks (lms, 2024) have predominantly focused on models that handle single or dual modalities, such as vision-language or video-text pairs, they fail to capture the complexities that arise when multiple modalities are involved. In real-world scenarios, user inputs are rarely confined to one or two modalities. Instead, they often consist of diverse combinations of text, images, videos, and audio, necessitating a holistic understanding and reasoning across information presented in these modalities for OLMs to effectively perform tasks. This mismatch between existing evaluation methods and the multimodal capabilities of state-of-the-art OLMs has left a significant gap in the assessment of these models.

One common flaw in existing OLMs is their inconsistent behavior when presented with the same question in different modalities or mixtures of modalities. Figure 1 presents an example on the Gemini 1.5 Flash (Gemini-Team, 2024a) (similar behaviour also observed in other OLMs, see Section 3.2 for analysis). Particularly, when the same math question is presented different modalities, such as rendered as image input, or spoke out as audio input, the model produces varying responses that exhibit significant performance discrepancies, *i.e.*, different reasoning bevhiours or different answers.

---

*Correspondence to Lichang Chen, `bobchen@umd.edu`, Hexiang Hu, `hexiang@google.com`, and Boqing Gong, `bgong@google.com`.

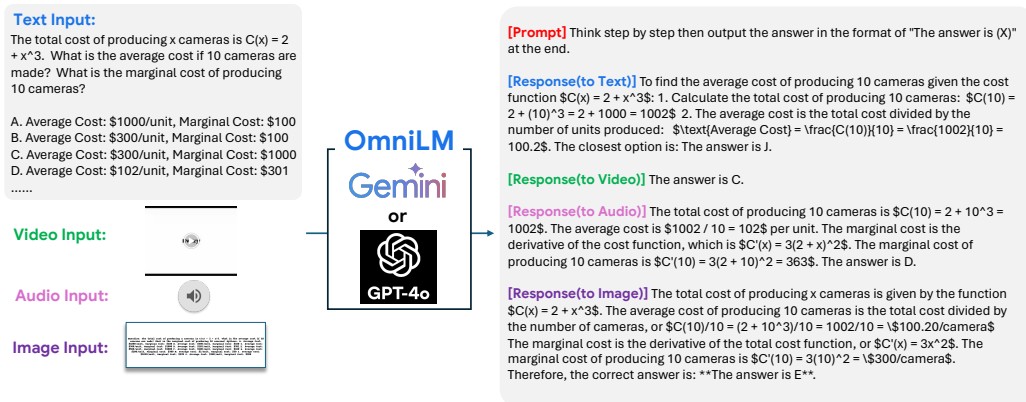

Figure 1: **Reasoning Behavior of a OLM Varies across Modalities**. Taking Gemini-1.5-Flash as an example, on text question, the reasoning behaviour is expected and the answer is correct. When the same question is rendered to an image, the model generate a reasonable reasoning but incorrect answer. On the video or audio representation of the same question, the model generates no reasoning and produces incorrect answers.

This observation indicates a lack of robust cross-modal information integration and reasoning capabilities in existing OLMs. Such inconsistency not only undermines the reliability of these models but also highlights the limitations of current evaluation benchmarks that do not adequately assess performance across diverse modality combinations.

To bridge this critical evaluation gap, we introduce Omni×R, an evaluation suite specifically designed to benchmark the reasoning performance of OLMs across a wide range of modalities. Unlike existing benchmarks that are limited to a maximum of two modalities, Omni×R provides a comprehensive testbed that includes complex modality combinations such as `video + audio + text` and `image + audio + text`, offering a more rigorous and holistic evaluation of these models' capabilities. Specifically, Omni×R contains two subsets of the data:

- **Omni×R$_{\text{SYNTH}}$**: a synthetic reasoning dataset constructed with a scalable and low-cost automatic method (*i.e.*, `Omnify!`) to translate information embedded in text to various modalities — audio, images, video, and hybrids of them.

- **Omni×R$_{\text{REAL}}$**: a real-world reasoning dataset manually collected and annotated with expert annotators, for evaluating cross-modal reasoning in the realistic distribution.

In construction of Omni×R$_{\text{SYNTH}}$, `Omnify!` translates text-based inputs into various other modalities, such as images, audio, and video, as well as their hybrid combinations, using programmatic text rendering services, programmatic video construction pipeline, and state-of-the-art text-to-speech service. This scalable synthetic dataset ensures a diverse and robust dataset that challenges OLMs to demonstrate their cross-modal reasoning abilities. Meanwhile, Omni×R$_{\text{REAL}}$ develops a realistic test environment for evaluating omni-modal reasoning. Particularly, we crawled 100 YouTube videos whose topics are related to math, physics, chemistry and coding, and manually curate, convert and annotate the quiz questions from those videos, ensuring that each question is associated with multiple variants, each in one modality or a hybrid of many modalities. With both complementary subsets, Omni×R allows us to better assess how well OLMs can reason across different modalities and integrate information in a way that mirrors human-like understanding.

Our evaluation of state-of-the-art OLMs on Omni×R has yielded several important findings. Notably, Omni×R is the first benchmark that quantitatively measured the aforementioned omni-modal behaviour discrepancy, especially in scenarios requiring deep reasoning across multiple modalities. Moreover, we also observe that some simple prompting strategy that exploits the underlying data creation logic in Omni×R$_{\text{SYNTH}}$, *i.e.*, `Extract the information and Then Answer` (ETA prompting), could significantly improve every omini-modality language model's behaviour consistency and final answer accuracy on Omni×R$_{\text{SYNTH}}$. These results suggest that the main struggle of current model is to establish a holistic understanding across modality, where the need to integrate and reason across different forms of data becomes crucial. When evaluated on Omni×R$_{\text{REAL}}$, where the information across modality is naturally distributed and blended with noises, OLMs can no longer rely on a simple prompting strategy to alleviate the omnimodal behaviour inconsistency, indicating the demand of further training for future omnimodal language models.

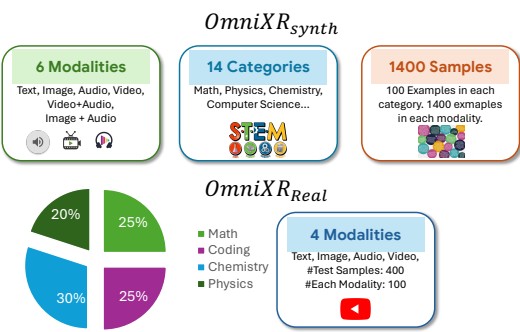

Figure 3: We propose `Omnify!` to create the synthetic omni-modality evaluation data from the original text benchmarks.

## 2 OMNI×R BENCHMARK

In this section, we introduce `Omnify!` a scalable and low-cost automatic method designed to translate text into various modalities, including audio, image, video, and combinations thereof. The overarching goal of `Omnify!` is to build up a scalable method to generate omni-modality data while keeping information the same across them for evaluating OLMs' reasoning capabilities across modalities. We construct the Omni×R benchmark in two subsets: (1) Omni×R_SYNTH: a synthetic omni-modal reasoning evaluation dataset derived from applying `Omnify!` on the MMLU-Pro (Wang et al., 2024). (2) Omni×R_REAL: a real-world omni-modal reasoning evaluation derived from Youtube, which is then processed and annotated by human experts.

### 2.1 OMNIFY!

**Text to image.** Though there are many ways to convert text into images, like using image generation models (e.g., Imagen-3 (Baldridge et al., 2024), DALLE-3 (OpenAI, 2024a)), however, the seemingly appealing text-to-image generation models make it challenging to control quality; they cannot ensure the generation contains all the information we need to answer a question. Before figuring out how to judge the quality of and information in the generated images, it is not viable to use image generators to scale up the mapping from text to images. Since our main goal is to evaluate models' reasoning capability, we start from the simplest approach in this work: rendering a canvas and then write the words on it. Given the images as input, we expect the models can achieve the same performance as they read text in this ideal scenario, where no extra noises, information losses, or variations are introduced by the text-to-image mapping process. Specifically, we use PIL[1] to create a new image with a white background and the text is drawn onto the image with black color. The engineering details/efforts can be found in Appendix I.

Figure 2: The overview of Omni×R_SYNTH and Omni×R_REAL.

**Text to Audio** We initially attempted to use Google Text-to-Speech[2] (TTS) for text-to-audio conversion. However, we encountered challenges with the mathematical equations. To address this, we developed a two-step process. First, we convert the original text, if it contains mathematical equations, into a format that is easy to speak orally. The details for the conversion could be found in Table 7. Then, we use a TTS engine to generate the audio, which contains the full information of the original text question.

**Text to Video** Like text-to-image generation models, there exist Sora (Brooks et al., 2024) and Veo (Google, 2024) we could leverage to map text to videos. However, they would incur the same problems as described in the text to image: quality control, time consumption, and computational cost. The main objective with videos here is to evaluate a model's capabilities on understanding a video input, which is a series of images from a model's view, and then reasoning to solve the problems. We fulfill this objective again using a simple approach to generating the video data from text as follows. Based on our image generation process, we render a series of images where each

---

[1] https://github.com/python-pillow/Pillow
[2] https://cloud.google.com/text-to-speech?hl=en

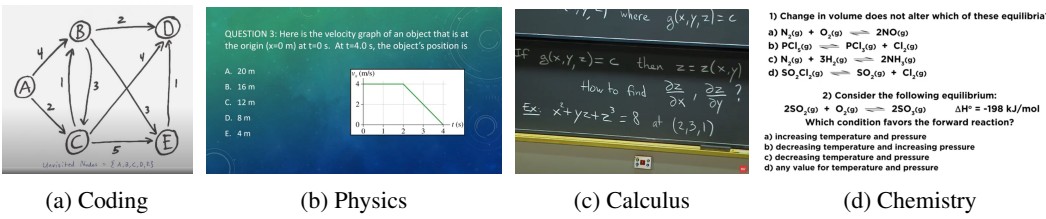

| (a) Coding | (b) Physics | (c) Calculus | (d) Chemistry |

Figure 4: Visualization of Examples in the Omni×R$_{\text{REAL}}$ set.

image contains one or several words from the text. We ensure that the information in the text is fully translated to the video. The input text is split into individual words first. Then we use OpenCV to create a video writer object with a specified frame rate, i.e., 1 FPS, and frame size (300x100 pixels). Each word is converted into an image using the text-to-image method. Finally, these images are combined sequentially to create video frames.

## 2.2 OMNI×R$_{\text{SYNTH}}$: SCALABLE SYNTHETIC OMINI-MODAL REASONING EVALUATION

Our initial choices of the text benchmark for `Omnify!` are Arc-Challenge (Clark et al., 2018) and GSM8K (Cobbe et al., 2021), but we identify the potential data contamination problems on these two benchmarks as Gemini-1.5-pro (Gemini-Team, 2024a) can achieve over 99% on GSM8K (results are shown in Table 11). It is very likely that contaminated OLMs just capture the part of the information they need from the video/audio questions and use their 'memory' to give correct answers, which cannot reflect the actual reasoning ability of the models. Thus, we choose MMLU-Pro (Wang et al., 2024), which is augmented from MMLU with ten options per question and released in June after the Gemini-1.5-Pro-001[3] release, as the text benchmark to `Omnify!`. In this way, we minimize the contamination influence, enabling a more accurate study of OLMs' omni-reasoning. We randomly sample 100 questions from each of the 14 categories in MMLU-Pro to construct Omni×R$_{\text{SYNTH}}$. Some examples for Audio and Video modalities are available[4].

## 2.3 OMNI×R$_{\text{REAL}}$: HIGH-QUALITY REAL-WORLD OMINI-MODAL REASONING EVALUATION

We crawl the video data from youtube and then transcribe it into different modalities to develop a realistic set as a valuable addition to the Omni×R.

**Video:** We select four categories that require dense reasoning in real-world scenarios: Mathematics, Coding, Physics, and Chemistry. Videos are sourced from popular educational channels, such as MIT OpenCourse. Two human annotators, spend approximately 30 hours each to review 100 videos (200 in total) and identify those containing non-trivial questions that demand substantial reasoning to solve. From these, 100 videos are carefully selected to construct a high-quality set, Omni×R$_{\text{REAL}}$. Each video clip is curated based on the following criteria: (1) it must contain one or more key frames that provide all the necessary information to solve the question; (2) the clip should exclude the answer to maintain the challenge; (3) some misleading or irrelevant frames are intentionally included to assess the model's robustness in reasoning. **Image:** We manually find the key frame(s) which contain the question information. It should be noted that in some cases, there might be several frames containing the relevant information, where we will crawl two or three frames and merge them together into one image. **Text:** Five human annotators transcribe the text from the video with the help of the tools, e.g., Gemini. All the open-ended generation questions are transferred into multiple choice questions to make the benchmark easy-to-use. **Audio:** The original audio will be checked first, which is extracted from the video we crawled. If it contains all the information for OLMs to answer the question, then we will just keep and use it. However, there are many cases where the audio does not contain the enough information for answering the questions, e.g., the instructor shows a slide and asks "solve the problems in the slide", where the problem is shown in image. In that scenario, we will use the same method in `Omnify!` to transfer the transribed text into audio by Google TTS.

---

[3] https://cloud.google.com/vertex-ai/generative-ai/docs/learn/model-versions
[4] https://anonymous.4open.science/r/OmnixR-Examples-7961/

## 3 Experiments and Findings

### 3.1 Experiment Setup

**Models.** We mainly test three series of models: Gemini (Gemini-Team, 2024a), *i.e.*, Gemini-1.5-Pro, and Gemini-1.5-Flash, OpenAI-GPT (OpenAI, 2024c), *i.e.*, GPT-4o and GPT-4o-mini, Anthropic-Claude (Anthropic, 2024), *i.e.*, Claude-3-Opus, Claude-3-Sonnet, Claude-3-Haiku. More details about the test models are shown in Appendix E.

**CoT Prompting.** The standard setting in MMLU-Pro (Wang et al., 2024) is to use Chain-of-Thought(CoT) prompting to elicit the reasoning ability of the OLMs for a more comprehensive evaluations. Following them, we use CoT with 0-shot, as our standard setting, *i.e.*, the prompt used for evaluation is "Think step by step then output the answer in the format of "The answer is (X)" at the end."

**Extract-Then-Answer (ETA) Prompting.** In addition, we employ Extract-Then-Answer (ETA) prompting, leveraging the benchmark's inherent structure. This method involves first extracting the textual content and then using the OLMs' language capabilities for reasoning to provide answers based on the transcriptions. To prevent potential hackings on Omni×R, we transparently demonstrate this approach in our benchmark, aiming for a comprehensive evaluation of OLMs. Specifically, the prompt 'Please extract the text from image/audio/videos' instructs the OLMs to function as text extractors. The extracted text from this initial step is subsequently fed back into the same OLM with Chain-of-Thought (CoT) prompting to obtain the final answer. Consequently, the model's performance reflects two key abilities: OCR/Transcription and Text Reasoning."

**Video/Audio/Image.** We first process the video to 1-fps to meet the requirements for both the Gemini and GPT models. For testing with Claude, we used the API available before August 10th, which only supported a maximum of 5 image inputs, so video evaluations were not conducted. The GPT-4o API supports 250 images input at the maximum, so any additional frames were dropped in the evaluation. In contrast, Gemini had no issues with the video modality and could handle all frames as input. Image processing is the modality that all models support most effectively, allowing comprehensive testing across all OLMs. Notably, Gemini is the only model supporting audio input.

**Answer Extraction:** We use the model to extract the answers. Since the regex parsing may affect the performance, we sacrifice the API cost to trade in the excellent extraction.

Table 1: **Results on Omni×R$_{\text{SYNTH}}$** show different mixed modalities evaluations, including text, image, audio, video. Each modality (Image/Audio/Video) combines two input sources: the 'Question' provided by the respective image, audio, or video modality, and the 'CoT instruction' provided by the text The numbers in red font, following the downward arrows, shows the drops compared to the pure text input.

| | Gemini 1.5 | | | | Claude | | | | | | GPT | | | |
| | Pro | | Flash | | Opus | | Sonnet | | Haiku | | 4o | | 4o-mini | |
| Modality | Perf. | Δ | Perf. | Δ | Perf. | Δ | Perf. | Δ | Perf. | Δ | Perf. | Δ | Perf. | Δ |
|---|---|---|---|---|---|---|---|---|---|---|---|---|---|---|
| Text | 77.5 | - | 69.9 | - | 77.7 | - | 77.4 | - | 72.5 | - | 71.5 | - | 72.6 | - |
| Image | 57.3 | 20.2↓ | 36.3 | 33.6↓ | 26.9 | 50.8↓ | 18.8 | 58.6↓ | 9.9 | 62.6↓ | 60.1 | 11.4↓ | 48.5 | 24.1↓ |
| Audio | 56.6 | 20.9↓ | 53.9 | 16.0↓ | - | - | - | - | - | - | - | - | - | - |
| Video | 36.3 | 41.2↓ | 15.1 | 54.8↓ | - | - | - | - | - | - | 53.1 | 18.4↓ | 18.6 | 54.0↓ |
| **Extract-Then-Answer (ETA) Prompting** | | | | | | | | | | | | | | |
| Image | 73.5 | 4.0↓ | 68.1 | 1.8↓ | 62.6 | 15.1↓ | 48.1 | 29.3↓ | 43.2 | 29.3↓ | 66.7 | 4.8↓ | 58.4 | 14.2↓ |
| Audio | 69.9 | 7.6↓ | 63.6 | 6.3↓ | - | - | - | - | - | - | - | - | - | - |
| Video | 48.6 | 28.9↓ | 42.8 | 27.1↓ | - | - | - | - | - | - | 25.0 | 46.5↓ | 59.3 | 13.3↓ |

### 3.2 Main Results on Omni×R$_{\text{SYNTH}}$

We show the main experimental results on ominified MMLU-Pro in Table 1.

Table 2: **Results on Omni×R$_{REAL}$** shows similar behaviour discrapancy of OLMs as indicated in results on the Omni×R$_{SYNTH}$. Interestingly, we also observe that simple prompting strategy (ETA prompting) is not as effective as it was on Omni×R$_{SYNTH}$, possibly due to the natural noise and redundancy in real-world image, video, and audio data.

| | Gemini 1.5 | | | | Claude | | | | | | GPT | | | |
| | Pro | | Flash | | Opus | | Sonnet | | Haiku | | 4o | | 4o-mini | |
| Modality | Perf. | Δ | Perf. | Δ | Perf. | Δ | Perf. | Δ | Perf. | Δ | Perf. | Δ | Perf. | Δ |
|---|---|---|---|---|---|---|---|---|---|---|---|---|---|---|
| Text | 86 | - | 80 | - | 78 | - | 66 | - | 65 | - | 85 | - | 75 | - |
| Image | 78 | 8↓ | 65 | 15↓ | 41 | 34↓ | 39 | 27↓ | 33 | 8↓ | 79 | 6↓ | 63 | 12↓ |
| Audio | 71 | 15↓ | 64 | 14↓ | - | - | - | - | - | - | - | - | - | |
| Video | 64 | 22↓ | 53 | 27↓ | - | - | - | - | - | - | 73 | 12↓ | 66 | 9↓ |
| **Extract-Then-Answer (ETA) Prompting** | | | | | | | | | | | | | | |
| Image | 79 | 7↓ | 65 | 15↓ | 63 | 15↓ | 52 | 14↓ | 51 | 14↓ | 79 | 6↓ | 70 | 5↓ |
| Audio | 55 | 31↓ | 51 | 29↓ | - | - | - | - | - | - | - | - | - | - |
| Video | 71 | 15↓ | 73 | 7↓ | - | - | - | | - | - | 66 | 19↓ | 63 | 12↓ |

**Model Comparison.** Gemini-1.5-Pro demonstrates the most versatile performance across all modalities, showing results in text, image, audio, video tasks. Claude models struggle with image tasks and lack audio and video capabilities. GPT models show a balanced performance, with GPT-4o performing particularly well in direct image and video compare to Gemini and Claude. Generally, larger models outperform their smaller counterparts across modalities, *e.g.*, Pro > Flash, Opus > Haiku). But interestingly, GPT-4o-mini outperforms GPT-4o in text and video with ETA prompting. For video tasks using ETA prompting, GPT-4o's performance inconsistencies led us to examine the model's responses to the extraction, we found that in over 46.8% test samples, the detailed analysis can be found in Appendix H, GPT-series models cannot extract the text from video, which we identify as the primary cause for the significant performance drop compared to CoT prompting. Regarding the text modality, two possible explanations emerge: first, MMLU-Pro was released before GPT-4o-mini, suggesting that OAI might have optimized for it. Second, since our dataset uses a subset sampled from MMLU-Pro, inherent biases may have influenced the results.

**Modality Analysis.** Text is the most mature modality across all models, with consistently high scores (ranging from 69.9% to 77.7%). Image modality shows significant variability, with direct task performance ranging from 9.9% (Claude Haiku) to 60.1% (GPT-4o). However, ETA prompting on image generally improves performance for all models, particularly for Claude (e.g., Opus improves from 18.8% to 62.6%). The improvement justifies the inclusion of ETA prompting as a standard in our benchmark to prevent potential manipulation. Audio modality, only available for Gemini models, shows moderate performance with notable improvement via ETA prompting. Video modality presents the most challenges, especially for the small models, *i.e.*, Gemini-1.5-Flash, and GPT-4o-mini.

There are also additional results on Arc-Challenge and GSM8k benchmarks shown in Table 11 with different modality input, *i.e.*, text, image, audio, video. Though the models are likely to be data contaminated on these benchmarks, the performance drops are still significant on image/video/audio compared to the pure text.

## 3.3 MAIN RESULTS ON OMNI×R$_{REAL}$

The results on the realistic set generally align with those from the synthetic set, showing significant drops in performance across audio, image, and video tasks compared to the text. One difference here is that performance on video does not drop a large margin compared to that in the synthetic set. Though the video is noisy than it is in the synthetic data, we can still capture one key frame and answer the question according to that key frame which largely reduces the difficulties, compared to the synthetic scenario, if the model can find the main frame in the video. Another interesting finding is that ETA prompting does not consistently improve performance; for example, there are performance drops in audio tasks with ETA prompting compared to CoT on both Gemini-Flash and Gemini-Pro. These findings confirm that our synthetic set effectively simulates real-world scenarios in a scalable, cost-efficient way, serving as a valuable sanity check for OLMs' omni-modality reasoning capabilities.

**Key Takeaways.** We summarize the following interesting takeaways from our experiments:

1. Multi-modal capabilities vary significantly across models, with Gemini 1.5 Pro showing the most broad support and balanced performance across all modalities.
2. Gaps still exists on other modalities compared to the text modality even just in such easy perception test scenarios. Significant room for improvement exists in video processing across all models, presenting opportunities for future development.
3. ETA prompting generally improves performance on Omni×R$_{\text{SYNTH}}$ but OLMs can no longer solely rely on it for Omni×R$_{\text{REAL}}$, indicating the necessity of the further alignment on omni-modality.
4. There's a clear trade-off between model size and performance, but smaller models (e.g., GPT-4o-mini) can sometimes outperform larger counterparts in specific tasks.
5. Our Omni×R$_{\text{SYNTH}}$ could be a good simulating set for the real-world scenarios, as the results on Omni×R$_{\text{REAL}}$ match the results in the Omni×R$_{\text{SYNTH}}$.

## 4 MIXED MODALITIES

Table 3: The results of more complex mixed modalities on Omni×R$_{\text{SYNTH}}$. We use the $\Delta$ to denote the performance drops from the text modality.

| Input Modality | | Gemini-Pro | | Gemini-Flash | |
|---|---|---|---|---|---|
| Question | CoT Prompt | Perf. | $\Delta$ | Perf. | $\Delta$ |
| Text | Text | 77.5 | - | 69.9 | - |
| Text | Video | 76.1 | 1.4↓ | 66.8 | 3.1↓ |
| Text | Audio | 74.1 | 3.4↓ | 68.3 | 1.6↓ |
| Text | Image | 74.1 | 3.4↓ | 66.9 | 3.0↓ |
| Image + Audio | Text | 61.8 | 15.7↓ | 49.1 | 20.8↓ |
| Video + Audio | Text | 40.1 | 37.4↓ | 25.9 | 44.0↓ |

**Text to Mixed Modalities.** In addition to the types of the `Omnify!` described in Section 2.1, our method could also be applied to generating interleaved modalities to better simulate more complex real-world scenarios, where the information is included in different modalities and requires a model to reason across the modalities to solve a problem. For example, an instructor can write down an equation on the blackboard and say "compute the derivative" in a Calculus lecture. Scenarios like this example require a model to jointly use image perception and audio understanding process the question, reason across the visual and audio modalities, and then provide a response. Using our `Omnify!`, we seamlessly integrate different modalities and create test samples with interleaved modalities, *i.e.*, "Video + Audio", and "Image + Audio", to Omni×R$_{\text{SYNTH}}$, which captures a more authentic user experience where multiple senses are engaged simultaneously. To be specific, We transfer the question into video and all the options are transferred for Audio, to get the modality, "Video + Audio", while CoT prompting remains in text form to maintain the model's reasoning ability across different modalities.

**Transferring CoT prompt to other modalities.** All the CoT prompting is in text for all the previous test cases. Here, we convert the CoT prompt into different modalities while keeping the others, *i.e.*, questions and options in MMLU-Pro intact.

**Results.** As shown in Table 3, there is a noticeable decline in performance when transitioning from text to mixed-modality tasks. For example, both the Pro and Flash models perform significantly worse in the "Video + Audio" scenario, achieving scores of 40.1 and 25.9, respectively. This indicates that handling mixed modalities presents a significant challenge, likely due to the increased complexity of integrating video and audio information. For Audio/Image/Video CoT, the model generally treats these inputs as noise or irrelevant context, having minimal impact on the final results, as performance approaches that observed with text-based CoT. We focus on evaluating the Gemini-series models since only Gemini supports audio inputs.

## 5 ANALYSIS

### 5.1 OMNI-MODALITY REASONING BEHAVIOUR ANALYSIS

After investigating the responses, we find that in omni-modality cases, Gemini-1.5-Flash models can only output very short answers though prompted to CoT before giving the answers, which is quite different from the reasoning behaviour in the pure-text. An example in Figure 1 shows the different behaviours among modalities, which intrigues us to have a quantitative analysis of the reasoning paths. We write a simple regex, detecting if the model output starts with "the answer/response is (*.)", with the rule, the total number of words should be less than 40, to evaluating whether the models' output contain the reasoning path. The results are shown in Table 4.

Table 4: The percentage of the model outputs containing the reasoning paths on Omni×R$_{\text{SYNTH}}$.

| Path(%) | Gemini 1.5 | | Claude | | | GPT | |
|---|---|---|---|---|---|---|---|
| Modality | Pro | Flash | Sonnet | Opus | Haiku | 4o | 4o-mini |
| Text | 98.9 | 89.1 | 100 | 100 | 98.6 | 100 | 100 |
| Image | 93.2 | 54.3 | 100 | 100 | 72.8 | 100 | 100 |
| Video | 91.3 | 23.4 | - | - | - | 99.1 | 95.7 |
| Audio | 94.0 | 82.3 | - | - | - | - | - |

Our analysis reveals that smaller models tend to produce reasoning paths less frequently for image, video, and audio inputs. Notably, for complex modalities like video, Gemini-1.5-Flash generates reasoning paths for only 23.4% of test examples, substantially lower than Gemini-1.5-Pro. Among the modalities, audio inputs elicit reasoning paths most similarly to text, while video inputs show the lowest rate of reasoning path generation. GPT-series models demonstrate excellent performance in producing reasoning paths across available modalities. However, these results underscore the significant challenges remaining in cross-modal reasoning. Given that models are expected to exhibit reasoning abilities, they should ideally output reasoning paths consistently across all input modalities.

### 5.2 VISUAL/VIDEO FORMATS INFLUENCES PERCEPTION PRECISION

#### 5.2.1 IMAGE

We first analyze how formats affect the performance on images. We show images with two different text formats in Figure 5. The lower image has a compact format, where the options are not spaced out; instead, they are presented in a continuous, inline format separated by periods. Compared to it, each option in the upper image is listed separately, making it easy to read, with letters (A to J) clearly aligned before each option. The results of CoT and ETA prompting with two different formats of images are shown in Table 6. The overall trend here is that with better format, we could significantly improve the performance across all the tested models. ETA prompting also boosts the performance for the both formats in general. For all the other models, the performance can be significantly improved when comparing BF with ETA, only the GPT-4o being an outlier.

We further analyze transcription accuracy using the Character Error Rate (CER), a standard metric for assessing text recognition performance, especially in OCR tasks. A CER of 0 indicates perfect accuracy, with higher values reflecting more errors. Details of the CER calculation are provided in Appendix J, and results are shown in Table 5. The results reveal that GPT-4o's OCR performance is largely format-independent, whereas other models exhibit considerable format sensitivity, explaining the pronounced improvements seen with ETA prompting for all models except GPT-4o when format is enhanced.

#### 5.2.2 VIDEO

We create different types of videos, one word per frame, several words per frame, etc. Our ablations reveal that increasing the number of words per frame generally leads to improved performance for both Gemini-Flash and Gemini-Pro models under both testing promptings, CoT and ETA prompting. This trend suggests that providing more context within each frame aids in the models' understanding and processing of the video content and narrow the gaps between images and videos.

Table 5: The Character Error Rate, the metric for evaluating the OCR, of different models on two different formats images.

| | Gemini 1.5 | | Claude | | | GPT | |
| --- | --- | --- | --- | --- | --- | --- | --- |
| | Pro | Flash | Opus | Sonnet | Haiku | 4o | 4o-mini |
| Image | 0.11 | 0.10 | 0.19 | 0.28 | 0.34 | 0.11 | 0.12 |
| Better Image | 0.06 | 0.03 | 0.05 | 0.18 | 0.26 | 0.11 | 0.11 |

Question: The relatives of a group of pelicans from the same species that separated from each other because of an unsuccessful migration are reunited 150 years later and find that they are unable to produce offspring. This is an example of

Options: A. gene flow.
B. temporal isolation.
C. disruptive selection.
D. founder effect.
E. genetic drift.
F. sexual selection.
G. sympatric speciation.
H. habitat fragmentation.
I. bottleneck effect.
J. allopatric speciation

Question: The relatives of a group of pelicans from the same species that separated from each other because of an unsuccessful migration are reunited 150 years later and find that they are unable to produce offspring. This is an example of Options: A. gene flow. B. temporal isolation. C. disruptive selection. D. founder effect. E. genetic drift. F. sexual selection. G. sympatric speciation. H. habitat fragmentation. I. bottleneck effect. J. allopatric speciation

Figure 5: We include two figures to illustrate which is a better format image. The upper one is the image with better format. The lower one is the image with the original format.

## 6    RELATED WORK

**Large Foundational Models.** GPT-4o (OpenAI, 2024b), Gemini (Gemini-Team, 2024a) both claim their models having omni-modality capabilities, but actually OAI's model does not support audio(no audio access via APIs)/video(only 250 frames and the videos should be separated manually before feeding into the model) while Gemini can take very long videos and has good Audio support. Claude (Anthropic, 2024) can be viewed as a vision-language model (Bordes et al., 2024) since it has capabilites to take image but no audio or video support. There are also other open-sourced vision language models, but they are mostly supporting only two modalities, e.g., the vision-language

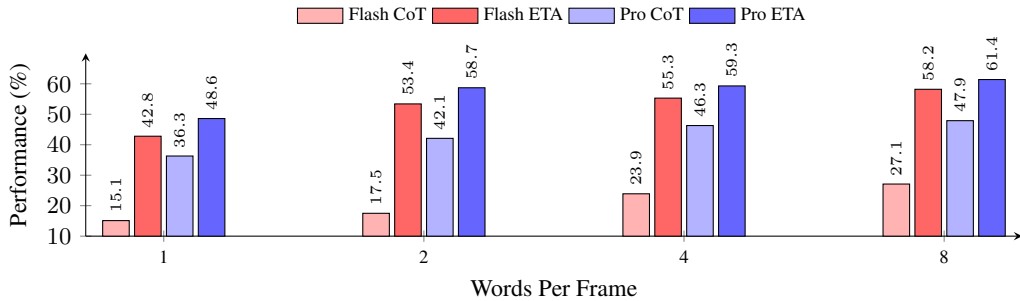

Figure 6: Video ablation study: Model performance with different words per frame. Pro and Flash denotes Gemini-1.5-Pro-001 and Gemini-1.5-Flash-001, respectively.

Table 6: The ablations: image with better format. BF: better format. The blue font denotes the performance gain of the better image compared to the original image format.

| | Prompt | Gemini 1.5 | | Claude | | | GPT | |
| | | Pro | Flash | Opus | Sonnet | Haiku | 4o | 4o-mini |
|---|---|---|---|---|---|---|---|---|
| Text | CoT | 77.5 | 69.9 | 77.7 | 77.4 | 76.5 | 71.5 | 72.6 |
| Image | CoT | 57.3 | 36.3 | 26.9 | 18.8 | 9.9 | 60.1 | 48.5 |
| Better Image | CoT | 64.6 7.3↑ | 43.6 7.3↑ | 33.5 6.6↑ | 28.9 10.1↑ | 19.1 9.2↑ | 65.5 5.4↑ | 52.1 3.6↑ |
| Image | ETA | 68.7 | 61.3 | 36.4 | 26.6 | 24.9 | 66.7 | 58.4 |
| Better Image | ETA | 73.5 4.8↑ | 68.1 6.8↑ | 62.6 26.2↑ | 48.1 21.5↑ | 43.2 18.3↑ | 66.9 0.2↑ | 61.7 3.3↑ |

models like LLaMA-3.1 and 3.2 (Meta, 2024), Pixtral (Mistral, 2024), LLaVA (Liu et al., 2023b;a); Audio-LLM like GAMA (Ghosh et al., 2024), LTU (Gong et al., 2023b;a), and SALMONN (Tang et al., 2024). It is hard to judge them on our benchmark, since the main idea behind our evaluations are that we expect the model has cross-modality reasoning and would like to encourage the model improving their cross-modal reasoning, only vision/audio/video would not get a comprehensive results. We would expect the open-sourced community to release real OLMs in the future and we will update the results accordingly.

**Video/Audio/Image Evaluation benchmarks.** Omnibench (Li et al., 2024b) specifically aimed at evaluating OLMs' tri-modal, i.e., text, vision, and audio, processing capabilities with human-annotated tasks. Compared to it, OmnixR emphasizes the omni-modality reasoning evaluations with both human-annotated realistic set and scalable synthetic set. MMMU (Yue et al., 2024a), MMMU-Pro (Yue et al., 2024b), CMMMU (Ge et al., 2024), and IsoBench (Fu et al., 2024) focus on evaluating vision-language models across various college-level disciplines with highly heterogeneous image types, emphasizing expert-level perception and reasoning across text-image pairs while LMSYS-Vision (lms, 2024) evaluates the instruction-following of the large vision-language models (Liu et al., 2023a; Chen et al., 2023; 2024; Yang et al., 2024a). Compared to them, OmnixR has larger scope on evaluating OLMs on cross-modality reasoning, not only vision input, but audio, video, and mixed modalities such as image + audio. AiShell-1, AiShell-2 (Du et al., 2018), Clotho-AQA (Lipping et al., 2022) are audio understanding benchmarks, providing extensive and high-quality real-world audio data for Mandarin ASR and audio question answering. MVBench (Li et al., 2024a) focuses on temporal reasoning across 20 challenging video tasks, Video-Bench (Ning et al., 2023) assesses Video-LLMs across video-exclusive, knowledge-based, and decision-making tasks, while MMBench-Video (Fang et al., 2024) offers a long-form, multi-shot evaluation of LVLMs with 609 videos and 2,000 human-annotated QA pairs across 26 fine-grained capabilities. In OmnixR, we also include long video in both synthetic and realistic scenarios and we also have mixed-modality evals including video + audio.

# 7 CONCLUSION

In this paper, we introduced Omnify!, a scalable and cost-efficient approach for generating multi-modal data from text, facilitating the construction of diverse and challenging test scenarios for omni-modal language models (OLMs). Using this method, we developed $Omni \times R_{SYNTH}$, a synthetic omni-modal reasoning evaluation dataset derived from MMLU-Pro, as well as $Omni \times R_{REAL}$, a real-world omni-modal reasoning dataset based on YouTube content. Our comprehensive evaluations reveal that OLMs experience substantial performance drops when confronted with complex multi-modal inputs, particularly in tasks that demand cross-modality reasoning. Notably, we observed that smaller models, *e.g.*, Gemini-1.5-Flash, are less adept at producing reasoning paths for image, video, and audio inputs compared to text, underscoring the inherent challenges in cross-modal reasoning. The evaluation results underscore the necessity for enhanced training strategies to address the complexities of omni-modal tasks. To sum up, $Omni \times R$ stands as a critical benchmark for guiding future advancements in OLMs, providing a foundation for measuring progress toward more human-aligned and truly omni-modal AI systems.

## 8 ACKNOWLEDGEMENT

We thank Quoc V. Le and Xinyun Chen for the discussions on the CoT reasoning and how LLM reasons with irrelevant context. We thank Prajit Ramachandran, Yiling Huang, Yichong Xu for insights and future works on synthetic audio evals/benchmarks. We thank Micheal Chang, Yao Fu on discussions of the real-world applications of Omni×R. We thank Jean-Baptiste Alayrac, Fangyu Liu on Helpful discussions for video/image evals and possibility to include the data into pretrain corpus. We thank Yong Cheng, Ruibo Liu, on helpful discussion on synthetic data vs. real-world data, and how to reduce data contamination on evaluations.

Lichang Chen and Heng Huang were partially supported by NSF IIS 2347592, 2348169, DBI 2405416, CCF 2348306, CNS 2347617.

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

## A  AUTHOR CONTRIBUTIONS

- **Lichang** devotes to revising the idea and constructing the Omni×R$_{\text{SYNTH}}$, finishing the code writing and most of the experiments.
- **Hexiang** proposed the initial idea for Omni-Eval via the observations on image reasoning behavior inconsistency of Gemini-Flash models.
- **Hexiang**, **Boqing**, **Yiwen**, **Zifeng**, **Mingda**, and **Yandong** contributed to the Omni×R$_{\text{REAL}}$.
- **Hexiang**, **Mingda**, **Yandong**, **Boqing**, and **Tianyi** attend the discussion regularly and provide useful feedback/suggestion for the project.
- **Pranav** contributes to the ablation study: images with better format and video ablations.
- **Heng** provided the university tuition support to Lichang.
- **Ming-Hsuan** and **Boqing** are project leads @Google Deepmind.

## B  CONVERT MATH INTO SPOKEN VERSION

For the math equations in the questions, we prompt Gemini-1.5-Pro to convert them into the version which can be spoken orally. The prompt we used is detailed in Table 7. We also show an example to explain the transformation: the TTS is hard to read the original question in Table 8 but it can handle the converted text.

Table 7: The oral conversion prompt designed for Text-to-Audio transfer.

| |
|---|
| [Prompt] Please transform all the equations in the text into the format that is easy to speak out orally. [Original text]
Please first output a single line of the text in the format "The transformed text is xxx" |

Table 8: An example of the conversion from the original question into the easily spoken text.

| |
|---|
| [Original Question] For what values of $x$ is it true that $x^2 - 5x - 4 \leq 10$? Express your answer in interval notation.
[Converted Text] The spoken version: For what values of x is x squared minus five x minus four less than or equal to ten? express your answer in interval notation. |

## C  RESULTS ON OPEN-SOURCED MODELS

We expanded our evaluation to include additional OLMs: Reka Flash (Team et al., 2024), Qwen-7B (Yang et al., 2024b), Qwen-72B (Yang et al., 2024a) and VideoLLaMA-2 (Cheng et al., 2024) on both the synthetic set (OmnixR-Synth) and the realistic set (OmnixR-Real). The results are shown in table 9 and table 10. The results of the three newly evaluated models align well with the findings in the paper:

1. Gaps in Modalities Beyond Text:Text modality consistently outperforms others across all models in both OmnixR-Synth and OmnixR-Real. For instance, on OmnixR-Synth, Reka-Flash achieves 62.5 in text but only 9.4 in image, 6.6 in video, and 16.3 in audio.
2. Significant Room for Improvement in Video: Video performance remains low across both synthetic and realistic datasets. For example, in OmnixR-Synth, the highest video score is 11.0 (Qwen-72B), and in OmnixR-Real, it is only 0.27.

## D  CATEGORIES IN MMLU-PRO

There are 14 categories in MMLU-Pro, including Math, Physics, Chemistry, Law, Engineering, Other, Economics, Health, History, Psychology, Business, Biology, Philosophy, Computer Science.

Table 9: **Open-sourced models on OmnixR-Synth.** We benchmark open-sourced models on our synthetic subset.

| Modality | Reka-Flash | Qwen 7B | Qwen 72B | VideoLLaMA2 |
|----------|-----------|---------|----------|-------------|
| Text     | 62.5      | 46.5    | 70.1     | 45.2        |
| Image    | 9.4       | 38.8    | 63.5     | 4.1         |
| Video    | 6.6       | 7.2     | 11.0     | 3.9         |
| Audio    | 16.3      | -       | -        | -           |

Table 10: **Open-sourced models on OmnixR-Real.** We benchmark open-sourced models on our realistic subset.

| Modality | Reka-Flash | Qwen 7B | Qwen 72B | VideoLLaMA2 |
|----------|-----------|---------|----------|-------------|
| Text     | 0.66      | 0.58    | 0.79     | 0.52        |
| Image    | 0.30      | 0.47    | 0.52     | 0.19        |
| Video    | 0.19      | 0.22    | 0.27     | 0.15        |
| Audio    | 0.23      | -       | -        | -           |

## E  MODEL SETTINGS/DETAILS

The version of the Geminis we used in this paper are Gemini-1.5-Pro-001 and Gemini-1.5-Flash-001. The version of the OpenAI models we used are gpt-4o-2024-05-13, and gpt-4o-mini-2024-07-18. The verison of the Claude models we used are claude-3-sonnet@20240229, claude-3-opus@20240229, claude-3-haiku@20240307.

The Gemini safety settings we used for video, audio, and images are shown in the following:

```
# Safety Setting
generative_models.SafetySetting(
    category=generative_models.HarmCategory.
        HARM_CATEGORY_DANGEROUS_CONTENT,
    threshold=generative_models.HarmBlockThreshold.BLOCK_ONLY_HIGH,
),
generative_models.SafetySetting(
    category=generative_models.HarmCategory.HARM_CATEGORY_HARASSMENT,
    threshold=generative_models.HarmBlockThreshold.BLOCK_ONLY_HIGH,
),
generative_models.SafetySetting(
    category=generative_models.HarmCategory.HARM_CATEGORY_HATE_SPEECH,
    threshold=generative_models.HarmBlockThreshold.BLOCK_ONLY_HIGH,
),
generative_models.SafetySetting(
    category=generative_models.HarmCategory.
        HARM_CATEGORY_SEXUALLY_EXPLICIT,
    threshold=generative_models.HarmBlockThreshold.BLOCK_ONLY_HIGH,
),
```

BLOCK_ONLY_HIGH is the loosest setting we can use for public Gemini APIs for video, audio, and images. BLOCK_ONLY_NONE is the loosest setting we can use for text, so we change all the Safety Settings for language into BLOCK_ONLY_NONE.

For response generation, we follow the commonly used settings, temperature=0.7, top_p=0.9, and output_length=1024, for all the models, *i.e.*, Gemini, Claude, GPT models.

## F  RESULTS ON ARC-CHALLENGE & GSM8K

We also evaluate Gemini models on ARC-Challenge dataset and GSM8K test set. The results are shown in Table 11.

Table 11: Performance of Gemini Models Across Different Modalities on ARC-Challenge and GSM8K Benchmarks

| Benchmark | Accuracy (%) | |
| --- | --- | --- |
| | Gemini-1.5-Pro | Gemini-1.5-Flash |
| **ARC-Challenge** | | |
| Text | 95.5 | 92.3 |
| Image | 79.5 | 75.0 |
| Audio | 91.1 | 88.0 |
| Video | 63.6 | 40.3 |
| **GSM8K** | | |
| Text | 99.1 | 96.3 |
| Image | 92.5 | 87.9 |
| Audio | 86.8 | 90.8 |
| Video | 80.3 | 63.1 |

## G   OMNI×R STATISTICS

We show the statistics of Omni×R$_{\text{REAL}}$ in table 12.

Table 12: Statistics for Video and Audio on the Omni×R$_{\text{SYNTH}}$. F: Frames, s: seconds.

| | Min | Max | Mean |
| --- | --- | --- | --- |
| Video | 28F | 552F | 117.2F |
| Audio | 7.2s | 251.3s | 32.3s |

Table 13: Statistics for Video and Audio on the Omni×R$_{\text{REAL}}$. F: Frames, s: seconds.

| | Min | Max | Mean |
| --- | --- | --- | --- |
| Video | 30f | 1326f | 255.6f |
| Audio | 10s | 1326s | 139.7s |

## H   ANALYZE THE EXTRACTION

We manually check the data first, and then find the patterns that the extraction failure have are mostly "unable to process", "can't extract", "I'm sorry", and "unable to extract". So we use these four patterns to check if the answers contain one of them, and calculate the percentage of the model answers which do not output the extractions when prompted as "Please extract the text from video."

## I    DETAILS OF THE TEXT-TO-IMAGE CONVERSION

We use the Python Imaging Library (PIL) to create a new image with a white background and the text is drawn onto the image with black color. The tricky part here is that the most commonly used font "times.ttf" does not support the Unicode well and will encounter the error when we try to convert the Unicode text, e.g., special mathematical symbols such as $\infty$, $\geq$, $\Pi$, $\Delta$. Thus, our solution here is to have a look-up-table to replace these Unicode text with latex code before generating. The details about the look-up-table is shown in Appendix I.1.

### I.1    LOOK-UP-TABLE FOR UNICODE CONVERSION

We show parts of look-up-table here due to the display issues. The full details about the look-up-table could be referred to our code.

```
'\u03b1': r'$\alpha$',   # Alpha
'\u03b2': r'$\beta$',    # Beta
'\u03b3': r'$\gamma$',   # Gamma
'\u03b4': r'$\delta$',   # Delta
'\u03c0': r'$\pi$',      # Pi
'\u03c3': r'$\sigma$',   # Sigma
'\u03c6': r'$\phi$',     # Phi
'\u03c9': r'$\omega$',   # Omega
'\u2211': r'$\sum$',     # Summation
'\u220f': r'$\prod$',    # Product
'\u222b': r'$\int$',     # Integral
'\u0394': r'$\Delta$',   # Capital Delta
'\u03a3': r'$\Sigma$',   # Capital Sigma
'\u03a6': r'$\Phi$',     # Capital Phi
'\u03a9': r'$\Omega$',   # Capital Omega
'\u2190': r'$\leftarrow$', # Left arrow
"\u2014": r"-",             # Em dash
"\u03a0": r"$\Pi$",        # Capital Pi
"\u0302": r"$^$",          # Circumflex accent
"\u2228": r"$\vee$",         # En dash
"\u2013": r"--",           # En dash
"\u2026": r"$\ldots$",        # Ellipsis
"\u2283": r"$\supset$",       # Superset
"\u201c": r"\"",         # Left double quotation mark
"\u03c1": r"$\rho$",       # Rho
"\u03b8": r"$\theta$",       # Theta
"\u03bb": r"$\lambda$",       # Lambda
"\u2225": r"$\parallel$",       # Parallel
"\u2022": r"$\\bullet$", # Bullet
```

## J    CER CALCULATION

The CER is calculated by

$$\text{CER} = \frac{S + I + D}{N}$$

Where $S$ = Number of substitutions, $I$ = Number of insertion, $D$ = Number of deletions, $N$ = Total number of characters in the ground truth text. Since the model output will contain some addition text, e.g., starting with "The text in the image reads:", which will affect the final results, we will detect the "question:" by regex to eliminate the influence by that irrelevant text.

Table 14: Examples of different reasoning behaviour across modalities on GPT-4o. We use red font to mark the key errors in the reasoning path. The correct answer here is F. GPT-4o only respond correctly on the text modality input.

---

[Question] Question: Find 4(3/4) % of $24.85. Options: A.$1.50 B.$1.10 C.$0.99 D.$1.25 E.$1.35 F.$1.18 G.$1.00 H.$0.93 I.$1.20 J.$1.05"

[Prompt] Think step by step then output the answer in the format of "The answer is (X)" at the end.

[Response(to Text)] To find 4 (3/4) % of $24.85, we will follow these steps:

1. Convert the mixed number 4 (3/4) into an improper fraction: - 4 (3/4) = 4 + (3/4) = (4 * 4/4) + (3/4) = (16/4) + (3/4) = 19/4.

2. Convert the fraction to a percentage: - (19/4) % = (19/4) * (1/100) = 19/400.

3. Now, calculate 19/400 of $24.85: - First, find the value of $24.85 * (19/400): - $24.85 * (19/400) = $24.85 * 0.0475 (since 19/400 = 0.0475).

4. Perform the multiplication: - $24.85 * 0.0475 = $1.179875.

5. Round to two decimal places: - $1.179875 rounds to $1.18.

Now, we check the options provided: A. $1.50 B. $1.10 C. $0.99 D. $1.25 E. $1.35 F. $1.18 G. $1.00 H. $0.93 I. $1.20 J. $1.05 The closest match is F. $1.18. Therefore, the answer is F.

[Response(to Video)] 1. The question is asking to find 4/3 of $24.85.

2. First, understand the mathematical expression: (4/3) * $24.85.

3. Calculate the multiplication: - Divide $24.85 by 3 = $24.85 / 3 = $8.2833 (approx) - Multiply the result by 4 = $8.2833 * 4 = $33.1333 (approx)

4. The closest option to $33.13 among the given options is $33.13. The answer is $33.13.

[Response(to Image)] To find 3/4 of 24.85, follow these steps:

Calculate $\frac{3}{4}$ of 24.85: $\frac{3}{4} \times 24.85 = \frac{3 \times 24.85}{4}$, $3 \times 24.85 = 74.55$, Now divide by 4: $\frac{74.55}{4} = 18.6375$, Round to the nearest cent: This gives us approximately $18.64$. However, since we need to find $\frac{4}{4}$ of that value, we can directly multiply: $\frac{3}{4} \times 24.85 = 18.6375$, None of these options seem to match the calculated value directly. The final answer considering the options is 'The answer is not found among the options provided.'

---

