# OpenReview forum: "OmnixR: Evaluating Omni-modality Language Models on Reasoning across Modalities"
_ICLR.cc/2025/Conference — ICLR 2025 Poster_

### Official Review · Reviewer_JM5H · 2024-10-31

**Soundness:** 3
**Presentation:** 4
**Contribution:** 2
**Rating:** 6
**Confidence:** 3

**Summary:**

The paper introduces *Omni×R*, a benchmark suite for evaluating omni-modality language models (OLMs) like GPT-4o and Gemini. And its key contributions include:
1. **Datasets**: *Omni×RSYNTH*, a synthetic multi-modal dataset, and *Omni×RREAL*, a real-world, expert-curated dataset for cross-modal reasoning.
2.  **Pipeline**: It proposes *Omnify!* to create the synthetic omni-modality evaluation data from the original text benchmarks.
3. **Prompting Strategies**: Tests of "Extract-Then-Answer" (ETA) prompting indicate it helps with synthetic data but is less effective with real-world data.

*Omni×R* establishes a rigorous standard for assessing OLMs, advancing omni-modal AI development.

**Strengths:**

1. **Gap Addressed**: The benchmark addresses a crucial gap in evaluating omni-modality language models, offering a framework for comprehensive cross-modal reasoning that goes beyond single or dual-modality assessments.
s for advancing omni-modal AI systems.

2. **Methodological Strength**: The methodological approach is robust, with a well-structured benchmark that includes both synthetic and real-world datasets, along with clear evaluation protocols and analysis.

3. **Clarity and Organization**: The paper is well-organized, providing clear explanations and thoughtful insights into the challenges and current limitations of existing omni-modality language models.

**Weaknesses:**

Here are some weaknesses I observe:

1. **Motivation of the Benchmark**: I understand the authors' goal of establishing a unified and comprehensive multi-modal benchmark to test OLMs' capabilities for holistic understanding and reasoning across modalities. I think the *Omni×RREAL* dataset achieves this goal to some degree. However, for the *Omni×RSYNTH* dataset, I don’t think it effectively assesses an OLM's cross-modal reasoning capabilities. Since *Omni×RSYNTH* is derived by translating from pure text, the images and videos it contains are entirely text-based, which means it primarily tests OLMs' OCR abilities rather than other critical capabilities like temporal, spatial, and grounding capabilities. In real-world scenarios, when faced with this type of data (e.g., slides), it would be more practical to use expert models to extract text information from other modalities, and then pass the task to LLMs. A more meaningful scenario might involve using films, where audio and video from movies are input to OLMs to answer plot- and character-related questions. I believe this setup would provide a more valuable test for OLMs.


2. **Evaluation**: In the "Extract-Then-Answer" (ETA) prompting method, the authors first prompt the OLMs to generate corresponding OCR results and then answer questions based on the extracted text. As I mentioned above, given this approach, why do we even need OLMs for such tasks? We could just use expert models to handle the first step. In real-world scenarios, "Extract-Then-Answer" is impractical, so I feel that this evaluation method also lacks significance.

**Questions:**

Kindly find below additional questions.

1. Can you provide the results of video+audio on the *Omni×Real* dataset?

2. For Claude and GPT-4o, you only tested video and image or just the image modality. So, why not test more open-sourced models? like LLaVA, QWen, etc. They can receive videos and images as input.

3. Why not test some open-source OLMs, such as Macaw-LLM, Next-GPT, or VideoLLaMA2?

**Details Of Ethics Concerns:**

No Ethics Concerns

---

> ### Author Response · Authors · 2024-11-22
>
> Thanks for your review and constructive feedbacks. Below, we address each of the main concerns raised.
>
> **Q1:** Can you provide the results of video+audio on the Omni×Real dataset?
>
> **A1:**  Firstly, we would like to clarify that as detailed in our paper, in the OmnixR-Real subset, each modality—text, video, audio, or image—independently provides all the information necessary for Omni-modality Language Models (OLMs) to answer a question.
> However, to evaluate the performance of Gemini models based on your request, we concatenated both video and audio into a single query. The results are presented below:
> | Modality  | Gemini-Flash | Gemini-Pro |
> |---|---|---|
> | Video + Audio   | 0.65         | 0.73       |
>
> It is important to note that this "Video + Audio" setting differs from the configuration in OmnixR-Synth. In OmnixR-Synth, the query is divided into two parts: the "Question" is provided in video format, while the "Options" are presented in audio format.
>
> **Q2:**  Why not test more open-sourced models? like LLaVA, QWen, etc. They can receive videos and images as input.
> **A2:** We evaluate four more omni-LLMs, i.e., Reka, Qwen-7B, Qwen-72B, and  VideoLLaMA2 and provide the results in the A2 of the General Response.
>
> **Q3:** Concerns about the motivation for the Omni×RSYNTH benchmark: While the Omni×RREAL dataset effectively evaluates OLMs' holistic understanding and reasoning across modalities, the Omni×RSYNTH dataset seems to primarily test OCR capabilities since its images and videos are derived from pure text. A more meaningful test could involve using real-world multimodal data, such as films, to evaluate OLMs' temporal, spatial, and grounding abilities.
>
> **A3:** The Omni×RSYNTH benchmark was designed as a scalable and low-cost method to evaluate OLMs' reasoning across non-text modalities. Its key advantage lies in the ability to quickly generate test data from any new text benchmark using the "omnify" method, mitigating risks of data contamination and enabling flexible updates, unlike the time-consuming and costly process of collecting realistic multimodal data. While we acknowledge that Omni×RSYNTH has limitations compared to real-world scenarios, it serves as a valuable sanity check for OLMs (as noted in line 322 of our paper). If the OLMs failed on our benchmark, then it will also perform very bad on other benchmarks. Despite its synthetic nature, the dataset highlights significant performance gaps between text and other modalities like images, videos, and audio, making it a meaningful resource for the community to assess emerging OLMs.
>
> **Q4:** Concerns about Evaluation: In the "Extract-Then-Answer" (ETA) prompting method, the authors first prompt OLMs to generate OCR results and then answer questions based on the extracted text. Given this approach, why do we even need OLMs for such tasks? Expert models could handle the OCR step more effectively. In real-world scenarios, "Extract-Then-Answer" seems impractical, so this evaluation method feels less significant.
>
> **A4:** The ETA prompting method was introduced primarily to address concerns about benchmark hacking, which is a significant issue in the community. By using ETA, we aim to discourage potential exploitation of benchmark artifacts, as we observed that certain prompting strategies could inflate performance on synthetic datasets like Omni×RSYNTH. We encourage researchers to evaluate their OLMs using both CoT and ETA prompting to provide a more comprehensive assessment.
> Although expert models can handle specific tasks like OCR more efficiently, our goal is to develop truly generalist OLMs capable of seamlessly integrating OCR and reasoning within a single framework. The ETA prompting method is designed to test these integrated capabilities, including OCR and multimodal reasoning, to better understand the current limitations and strengths of OLMs.

---

> ### Author Response · Authors · 2024-11-25
>
> Dear Reviewer JM5H,
>
> As we are approaching the deadline of the discussion period, we would like to cordially inquire about the extent to which we have successfully addressed the concerns outlined in your review. Should there be any lingering points that require further attention, please rest assured that we are enthusiastic about the opportunity to provide comprehensive responses to any subsequent queries or comments you may have.
>
> Your constructive input remains invaluable to us, and we appreciate your dedication to enhancing the quality of our manuscript. Thank you for your time and consideration.
>
> Best,
>
> Authors

---

> ### Comment · Reviewer_JM5H · 2024-11-25
>
> Thank you for the feedback. I will improve my score, but I still hope the authors can incorporate more real-world testing examples into the benchmark. Additionally, I would like to point out that the significant performance gap between text and other modalities (such as images, videos, and audio) has been addressed in many previous works [1,2]. I believe this is a widely acknowledged fact in the field.
>
> [1] IsoBench: Benchmarking Multimodal Foundation Models on Isomorphic Representations
>
> [2] MATHVERSE: Does Your Multi-modal LLM Truly See the Diagrams in Visual Math Problems?

---

> ### Author Response · Authors · 2024-12-03
>
> Thank you for your feedback! We will continue to refine our work and incorporate more real-world examples into our benchmark to enhance its utility. We also greatly appreciate the references you provided and will ensure they are cited and discussed in the related work section of our paper. However, we would like to point out that these works primarily focus on the image-text modality, which we believe is insufficient for evaluating state-of-the-art omni-modality language models. Our benchmark aims to provide a more comprehensive evaluation across a broader range of modalities, including video and audio, to better capture the capabilities of these advanced models.

---

### Official Review · Reviewer_aEZD · 2024-11-03

**Soundness:** 4
**Presentation:** 3
**Contribution:** 3
**Rating:** 6
**Confidence:** 4

**Summary:**

This paper proposes a novel evaluation suite called Omni x R for omni-modality language models (OLMs) evaluation. They propose an automatic data synthetic method to convert natural language questions into other modalities, i.e., audio, image, video. Besides, they also annotate 400 samples from real videos to  other modalities. With Omni x R, they conduct experiments of several OLMs and provide insights to the develop of future OLMs

**Strengths:**

1. This paper proposed a novel evaluating method called Omni x R for Omni-modality Language Models (OLMs) to assess OLMs over a diverse mix of modalities. Unlike other benchmarks that focus on one or two modalities, Omni x R evolves up to six modalities that embed information, presenting new challenges for OLMs.
2. Omni x R can effectively measure the reasoning behavior discrepancy in different modalities. Besides, this paper demonstrates several key findings regarding this discrepancy. For example, they find extracting information and then conducting reasoning could improve model performance on modalities other than text.

**Weaknesses:**

1. The automatic data synthesis method is somehow oversimplified. For example, rendering text onto a white background image or using tts to translate text to audio may be used in synthetic training data generation. These kinds of data could vary largely from the data distribution in the real world, which makes the OMNI-R-Synth data unreliable in reflecting the model performance in real life.
2. The benchmark size is small. There are only 400 real samples and 400 synthetic samples. Besides, Demonstrations of some examples of OMNI-R-Synth and OMNI-R-Real could help readers gain an intuitive impression of the data format.
3. I’m curious about how humans perform in different modalities. A hypnosis is that audio could cause difficulty for humans to perform reasoning. Are the authors planning to add a human upper bound for Omni x R?
4. When conducting ETA prompting, did the author measure the precision of the intermediate textual transcripts? Does the performance drop come from failure to catch the key information or failure to conduct reasoning when some modalities exist?

**Questions:**

Please see above.

---

> ### Author Response · Authors · 2024-11-22
>
> Thanks for your review and constructive feedbacks. Below, we address each of the main concerns raised.
>
>
> **Q1:** The benchmark is small. There are only 400 real samples and 400 synthetic samples.
>
> **A1:** We would like to clarify that our synthetic set contains 1,400 samples per modality, amounting to 5,600 samples across four modalities. Additionally, our benchmark includes mixed-modality samples, such as image+audio and video+audio, with 1,400 samples for each combination. Altogether, the benchmark comprises 8,400 synthetic samples alongside 400 real examples. Given the high computational and resource costs associated with evaluating multimodal data, we believe this is a reasonable and practical benchmark size.
>
> **Q2:** The demonstrations of the data
>
> **A2:** Actually, since we cannot write the video/audio into our paper, we provide one anonymous link at the bottom of the Page4: https://anonymous.4open.science/r/OmnixR-Examples-7961/ and claim we uploaded some examples to this link in Line188. We add more examples on Anonymous Github: https://anonymous.4open.science/r/example-omnixr-7C6E/.
>
>
> **Q3:** I’m curious about how humans perform in different modalities. A hypnosis is that audio could cause difficulty for humans to perform reasoning. Are the authors planning to add a human upper bound for Omni x R?
>
> **A3:** Thanks for your feedback! These are all very difficult questions for human beings since they are very complex questions. For example, there is one extremely difficult category called Abstract Algebra, which evaluates models on topics typically covered in advanced undergraduate or graduate-level mathematics courses, including Group Theory, Ring and Field Theory. Asking a non-expert human to answer is almost infeasible, especially in other modalites, e.g., audio. In the original MMLU-Pro paper, they also do not have the human upper bound. Thus, we would like to keep the same setting.
>
> **Q4:** When conducting ETA prompting, did the author measure the precision of the intermediate textual transcripts? Does the performance drop come from failure to catch the key information or failure to conduct reasoning when some modalities exist?
>
> **A4:** Thanks for your questions! For ETA prompting, the failures mainly come from the perception/ASR errors. The models tested in the paper can do the reasoning on the text modality quite well. We also have the error analysis of the direct CoT testing shown in Section 5.1, where we show that the reasoning behavior is quite different among different modalities, e.g., Gemini-Flash is hard to output the reasoning paths on image and video modalities though they are prompted to “think step by step before giving the answer”.

---

> ### Author Response · Authors · 2024-11-25
>
> Dear Reviewer aEZD,
>
> As we are approaching the deadline of the discussion period, we would like to cordially inquire about the extent to which we have successfully addressed the concerns outlined in your review. Should there be any lingering points that require further attention, please rest assured that we are enthusiastic about the opportunity to provide comprehensive responses to any subsequent queries or comments you may have.
>
> Your constructive input remains invaluable to us, and we appreciate your dedication to enhancing the quality of our manuscript. Thank you for your time and consideration.
>
> Best,
>
> Authors

---

### Official Review · Reviewer_5iEv · 2024-11-03

**Soundness:** 3
**Presentation:** 3
**Contribution:** 3
**Rating:** 6
**Confidence:** 4

**Summary:**

Authors propose the Omni x R benchmark intended to evaluate Omni LLM, presenting three modalities: text, vision and audio. Two variants are proposed: Omni x R_synth, which is synthetic built, and Omni x R_real that has been human curated. The sources for those subsets are respectively MMLU-Pro and Youtube.

Along those benchmarks, the authors introduce the Omnify method, which offers to create synthetic omni-modality evaluation data by 'translating' across modalities data sample (e.g. video to image and audio, or audio and text, etc).

**Strengths:**

- The paper is well written. Data sources, models and experiments are clearly stated with examples. Takeaways are given in page 7, that help the reader understanding the scope of the contributions.
- Good point on the authors to spot the data contamination with Gemini on GSM8K.

**Weaknesses:**

- One big weakness of this work seems that the whole paper doesn't show examples of Omni x R_synth or Omni x R_real. It is unclear for the readers how exactly those benchmark look like in reality? One could assume that Omni x R_synth is probably a MCQ-type of benchmark, but this isn't stated in the paper. What Omni x R_real is looking in practice is even less clear in those settings. As those two benchmarks are the main contribution of this work, it is a bit unclear why the authors have not shown them in first place. Arguably Figure 1 should be not-cherry-picked examples of Omni x R_synth and Omni x R_real.
- The current Figure 1 is not very informative. The source of that example isn't mentioned. The terminology "OmniLM" in the figure seems to conflict with the rest of the paper ("omni-LM"?). The figure mentions "Gemini or GPT-4o". The 'OR' alone is misleading. There are actually many omni-LLM in the community, starting with Reka-Core, UnifiedIO2, AnyGPT, MIO-Instruct, and many others. They all seem to be ignored by this paper. Finally, the legend mentions using 'Gemini'. Arguably, Figure 1 should be removed altogether and replaced with actual examples from Omni x R_synth and Omni x R_real.
- The omnify method doesn't seem to be evaluated.
- Terminology: The paper mixes freely "omnimodal" and "omni-modal". Please pick one and keep the terminology consistent.
- Terminology: Figure 2 uses "OmniXR_{real}" while the rest of the paper uses "Omni x R_{real}". Why not using the same consistent terminology?
- Nitpicks: L092 s/Youtube/YouTube/, L126 s/Youtube/YouTube/, L192 s/youtube/YouTube/. The only correct is L529.
- Figure 2: The color code is misleading. Blue seems to be 'Chemistry' in the pie chart. The YouTube source is blue too. However, in L197 it is mentioned that 4 categories are used: Math, Coding, Physics and Chemistry. Same remark for the string '14 Categories' with a blue box above. This figure 2 seems completely inaccurate. Use color code when needed and accurately.
- As an evaluation paper that proposes a benchmark, the model evaluated are quite lacking. The work focused on Gemini, Claude and GPT-4o, putting aside many omni-LLM (Reka, AnyGPT, Video-SALMONN, UnifiedIO2, etc).

**Questions:**

- What is the license for your benchmarks? It is probably safe to assume that MMLU-Pro can be used freely by the community. However, scraping YouTube videos might be restricting the use of Omni x R. What is the license? Does it allow Commercial use, if say another company wants to use that benchmark to evaluate their internal OLM?
- L204: Frame merging is mentioned. How realistic is that? Can you show a not-cherry-picked example?
- Is there are reason why the evaluation was limited to closed-source models (Gemini, Claud and GPT-4o)? Why not evaluating the rest of the omni-LLM models?

---

> ### Author Response · Authors · 2024-11-22
>
> Great thanks for your constructive feedback and valuable insights. Below, we address each of the main concerns raised.
>
>
> **Q1:** The paper does not show examples.
>
> **A1:** We acknowledge the challenge of including video and audio examples directly in the paper. To address this, we provide an anonymous link at the bottom of Page 4 (https://anonymous.4open.science/r/OmnixR-Examples-7961/) where examples are uploaded, as noted in Line 188. Additionally, we have added more examples on another anonymous GitHub repository: https://anonymous.4open.science/r/example-omnixr-7C6E/.
>
> **Q2:** The current Figure 1 is not very informative. The source of that example isn't mentioned. The terminology "OmniLM" in the figure seems to conflict with the rest of the paper ("omni-LM"?). The figure mentions "Gemini or GPT-4o". The 'OR' alone is misleading. There are actually many omni-LLM in the community, starting with Reka-Core, UnifiedIO2, AnyGPT, MIO-Instruct, and many others. They all seem to be ignored by this paper. Finally, the legend mentions using 'Gemini'. Arguably, Figure 1 should be removed altogether and replaced with actual examples from Omni x R_synth and Omni x R_real.
>
> **A2:** Thanks for pointing out the problems in Figure 1. We will continue to refine our paper.
>
> **Q3:** The omnify method doesn't seem to be evaluated.
>
> **A3:** Thank you for your question! We conducted a detailed investigation of approximately 200 examples for each modality. While humans do not face issues in recognizing text from videos or images, challenges were observed with audio, particularly in transferring mathematical equations. To address this, we proposed a solution in our paper: converting mathematical equations into an easy-to-read format before applying the Omnify method, which is detailed in section 2.1 “text-to-audio” paragraph. An example of this conversion is provided in Table 8, demonstrating how the original question is transformed into a format suitable for spoken text. This version improves grammar, clarity, and flow while retaining the original content.
>
> **Q4:** Inconsistent Terminologies, typos.
>
> **A4:** Thanks for pointing out that. We will continue to polish the paper and fix the issues.
>
>
> **Q5:** As an evaluation paper that proposes a benchmark, the model evaluated are quite lacking. The work focused on Gemini, Claude and GPT-4o, putting aside many omni-LLM.
>
> **A5:** We evaluate four more omni-LLMs, i.e., Reka, Qwen-7B, Qwen-72B, and  VideoLLaMA2 and provide the results in the A2 of the General Response.
>
>
> **Q6:** What is the license for your benchmarks? It is probably safe to assume that MMLU-Pro can be used freely by the community. However, scraping YouTube videos might be restricting the use of Omni x R. What is the license? Does it allow Commercial use, if say another company wants to use that benchmark to evaluate their internal OLM?
>
> **A6:** Our benchmark dataset will inherit the licensing terms of its components. MMLU-Pro, as a freely available benchmark, can be used without restrictions, including for commercial purposes. For the OmnixR-Real subset, which includes annotations for selected YouTube videos, we will release the code for dataset curation to facilitate academic research. However, we do not claim ownership of the underlying videos, and their use is subject to YouTube’s Terms of Service and applicable copyright laws. While we will release all synthetic data, users must ensure compliance with relevant licensing and copyright requirements when using the dataset for any purpose, including commercial evaluation.
>
> **Q7:** L204: Frame merging is mentioned. How realistic is that? Can you show a not-cherry-picked example?
>
> **A7:** Thanks for your question. We provide the examples in the following anonymous link:  https://anonymous.4open.science/r/example-omnixr-7C6E/. It is a challenging task that the model should recognize every frame to answer the question since we put one word each frame.

---

> > ### Comment · Reviewer_5iEv · 2024-11-25
> >
> > Thank you for the feedback. While the effort to have examples on a separate server is appreciated, I don’t understand why those examples aren’t integrated directly into the main PDF? Understandably audio examples might require to have a transcription and video ones to have sampled frames. However since this is the main contribution of this work the presence of at least a very single one real example of the benchmark seems a mandatory addition. I should mention that this work isn’t the only Omni benchmark I have reviewed in this review cycle, and other works don’t seem to have an issue at clearly showing actual data points of the very benchmark proposed as contribution.

---

> > > ### Author Response · Authors · 2024-12-04
> > >
> > > Great thanks for your feedback! We will continue to refine our work and include the real examples which we show in the separated server into our paper in our camera-ready version!

---

### Official Review · Reviewer_89Zj · 2024-11-03

**Soundness:** 3
**Presentation:** 3
**Contribution:** 3
**Rating:** 6
**Confidence:** 4

**Summary:**

This paper proposes an evaluation suite for multi-modal language models - Omni×R, which can integrate various modalities such as text, vision, and audio. The design of Omni×R aims to assess the models' capabilities in cross-modal reasoning, especially when the user's messages contain multiple modalities, the models need to be able to understand and reason comprehensively to complete the task. The paper introduces two evaluation variants: synthetic data Omni×RSYNTH and real-world data Omni×RREAL. These evaluations reveal the existing problems of the state-of-the-art models.

**Strengths:**

- The motivation of this paper is very clear, the problem proposed is valid, and the benchmark proposed is distinctly different from others.
- The evaluation experiments in this paper are comprehensive, exploring the experimental results under different modality combinations and different prompt scenarios, and providing a detailed analysis.
- The pipeline of data construction is inspiring, providing a possible solution for the subsequent generation of cross-modal datasets.

**Weaknesses:**

- The synthesis method for audio data is common, but the direct rendering approach for images and videos seems overly simplistic and overly reliant on the model's OCR capabilities.
- While the paper points out the challenges models face in cross-modal tasks, it does not provide specific model optimization strategies or directions for improvement.

**Questions:**

- Why is the performance of ETA Prompting also poor in Table 2, especially on the audio modality where it performs worse than the normal evaluation?
- I would like to know if the authors have cleaned the constructed data?

---

> ### Author Response · Authors · 2024-11-21
>
> Great thanks for your constructive feedback and valuable insights. Below, we address each of the main concerns raised.
>
> **Q1:** Weakness on not providing the specific model optimization strategies or directions for improvement.
>
> **A1:** We have a prompting method to alleviate the performance degradation. But for model optimization, since our focus is building up a benchmark for evaluating the omni-reasoning, we leave the optimizing model’s performance for the future work.
>
> As for the directions for improvement, we believe we provide it in the design logics of our benchmark, i.e., the model should focus on not only the text reasoning. The small models should focus more on the multimodal reasoning, especially the CoT, since it has difficulties to output the reasoning paths.
>
> **Q2:** The synthesis method for audio data is common, but the direct rendering approach for images and videos seems overly simplistic and overly reliant on the model's OCR capabilities.
>
> **A2:** Firstly, we would like to argue that OCR is one ability model should have. Secondly, based on both the results in our paper, the gaps between image/video and text are still pretty large. It means that the model still has a large room to improve or climb up on our benchmark even though it is simple in image/video. The benchmark can serve as a good sanity check for the OLMs, i.e., if they cannot achieve good performance on our benchmark, when it comes to the more complex real-world scenarios, they will fail as well.
>
> **Q3:** I would like to know if the authors have cleaned the constructed data?
>
> **A3:** For OmnixR-Real, we manually verified the entire dataset after the annotators completed their labeling process. Therefore, we consider it a thoroughly cleaned dataset.
>
> For OmnixR-Synth, we conducted a detailed analysis of approximately 200 examples for each modality. While recognizing text from videos or images posed no significant challenges for humans, difficulties arose with audio, particularly in transferring mathematical equations. To address this issue, we proposed a solution in Section 2.1 (under the "Text-to-Audio" paragraph) of our paper: converting mathematical equations into a more accessible format before applying the Omnify method. An example of this conversion can be found in Table 8.
>
> **Q4:** Why is the performance of ETA Prompting also poor in Table 2.
>
> **A4:** In real-world scenarios, relying solely on ETA prompting is insufficient for addressing complex problems. For example, in the case of complex videos, extracting information first can introduce irrelevant context, and key frames critical to solving the question might fail to be transcribed. The design rationale behind ETA prompting is to discourage exploiting shortcuts or "hacking" our benchmarks. Therefore, we do not intend for ETA prompting to serve as a method for improving model performance on our benchmarks but rather as an evaluation approach.

---

> > ### Comment · Reviewer_89Zj · 2024-11-25
> >
> > Thanks for your response. Most of the questions I raised have been well resolved, and I have updated the rating accordingly.

---

> > > ### Author Response · Authors · 2024-11-25
> > >
> > > Thank you for your prompt response and thanks for the suggestion! We will continue to polish our paper and make it better.

---

### Author Response · Authors · 2024-11-21
**General Response**

Firstly, we would like to thank all the reviewers for providing useful feedback. We will follow these helpful suggestions to continue polishing our paper and releasing the benchmark as soon as possible to contribute to the community. We summarize the concerns that reviewers have and give the general responses as follows.

**Q1:** The paper does not show examples. Could you please show more non-cherry-picked examples of OmnixR?

**A1:** We acknowledge the challenge of including video and audio examples directly in the paper. To address this, we provide an anonymous link at the bottom of Page 4 (https://anonymous.4open.science/r/OmnixR-Examples-7961/) where examples are uploaded, as noted in Line 188. Additionally, we have added more examples on another anonymous GitHub repository: https://anonymous.4open.science/r/example-omnixr-7C6E/.

**Q2:** Evaluate more Omni-modality Language Models~(OLMs), e.g., Reka, Qwen, VideoLLaMA-2.

**A2:** We expanded our evaluation to include additional OLMs—Reka Flash, Qwen-7B, Qwen-72B, and VideoLLaMA-2—on both the synthetic set (OmnixR-Synth) and the realistic set (OmnixR-Real). The results are summarized below:
### OmnixR-Synth
| Modality | Reka-Flash | Qwen 7B | Qwen 72B | VideoLLaMA2 |
|---|---|---|---|---|
| Text | 62.5 | 46.5 | 70.1 | 45.2 |
| Image | 9.4 | 38.8 | 63.5 | 4.1 |
| Video | 6.6 | 7.2 | 11.0 | 3.9 |
| Audio | 16.3 | - | - | - |

### OmnixR-Real
| Modality | Reka-Flash | Qwen 7B | Qwen 72B | VideoLLaMA2 |
|---|---|---|---|---|
| Text | 0.66 | 0.58 | 0.79 | 0.52 |
| Image | 0.30 | 0.47 | 0.52 | 0.19 |
| Video| 0.19 | 0.22 | 0.27 | 0.15 |
| Audio | 0.23 | - | - | - |

The results of the three newly evaluated models align well with the findings in the paper:
1. Gaps in Modalities Beyond Text:Text modality consistently outperforms others across all models in both OmnixR-Synth and OmnixR-Real. For instance, on OmnixR-Synth, Reka-Flash achieves 62.5 in text but only 9.4 in image, 6.6 in video, and 16.3 in audio.
2. Significant Room for Improvement in Video: Video performance remains low across both synthetic and realistic datasets. For example, in OmnixR-Synth, the highest video score is 11.0 (Qwen-72B), and in OmnixR-Real, it is only 0.27.

These trends, observed consistently across OmnixR-Synth and OmnixR-Real, validate the utility of the synthetic set in simulating real-world scenarios while highlighting the significant performance gaps in non-text modalities.

**Q3:** Evaluations of the omnify method.

**A3:** Thank you for your question! We conducted a detailed investigation of approximately 200 examples for each modality. While humans do not face issues in recognizing text from videos or images, challenges were observed with audio, particularly in transferring mathematical equations. To address this, we proposed a solution in our paper: converting mathematical equations into an easy-to-read format before applying the Omnify method, which is detailed in section 2.1 “text-to-audio” paragraph. An example of this conversion is provided in Table 8, demonstrating how the original question is transformed into a format suitable for spoken text. This version improves grammar, clarity, and flow while retaining the original content.

---

### Meta-Review · Area_Chair_Fqaw · 2024-12-18

**Metareview:**

This paper introduces OmnixR, a benchmark designed to evaluate omni-modality language models (OLMs) across text, vision, and audio domains. The authors clearly articulate the motivation and assemble both synthetic and real-world datasets to assess cross-modal reasoning. Reviewers commend the work's clarity, breadth of experiments, and the constructive evaluation pipeline. The authors have addressed initial concerns by providing more examples and extending evaluations to additional OLMs. Although some reviewers suggest that synthetic data primarily tests OCR-like skills and question the practical significance of certain prompting strategies, the consensus is that OmnixR offers a valuable resource to the community. The benchmark highlights key challenges in cross-modal reasoning and lays a foundation for future improvements. I recommend acceptance.

**Additional Comments On Reviewer Discussion:**

During the discussion phase, the authors provided more non-cherry-picked examples and evaluated additional OLMs, addressing primary reviewer concerns. Their commitment to refining the datasets and sharing them openly was well received. Considering these improvements and the benchmark's potential utility, the final decision is to accept the paper.

---

### Decision · Program_Chairs · 2025-01-22

Accept (Poster)